# Accuracy of Analog and Digital Full-Arch Mandibular Impressions: In Vitro and In Vivo Evaluation

**DOI:** 10.3390/diagnostics15162077

**Published:** 2025-08-19

**Authors:** Diana Cerghizan, Kinga Mária Jánosi, Alexandra Farcas, Marcel Mihai Bojan, Mircea Horia Muntean, Andreea Ana Maria Nechiti, Izabella Éva Mureșan, Silvia Izabella Pop, Gyula Marada

**Affiliations:** 1Faculty of Dental Medicine, George Emil Palade University of Medicine, Pharmacy, Science and Technology of Târgu Mureș, 38 Gh. Marinescu Str., 540139 Târgu Mureș, Romanianechiti_andreea@yahoo.com (A.A.M.N.);; 2Department of Isotopic and Molecular Technologies, National Institute for Research and Development of Iso-Topic and Molecular Technologies, 67-103 Donat Street, 400293 Cluj-Napoca, Romania; 3Manufacturing Engineering Department, Faculty of Industrial Engineering, Robotics and Production Management, Technical University of Cluj-Napoca, B-dul Muncii, 103-105, 400641 Cluj-Napoca, Romania; 4Department of Prosthodontics, Faculty of Medicine, University of Pécs, Tüzér Street 1., H-7625 Pécs, Hungary; marada.gyula@pte.hu

**Keywords:** accuracy, conventional impression, digital impression, intraoral scanner, transfer aid

## Abstract

**Background/Objectives:** Accurate full-arch impressions are crucial for predictable prosthodontic outcomes. While intraoral scanners (IOSs) are increasingly adopted, evidence comparing their accuracy with conventional analog impressions across full mandibular arches—particularly under both laboratory and clinical conditions using an objective intraoral reference—is limited. Our study aims to evaluate the in vitro and in vivo accuracy of digital impressions compared to conventional methods in full-arch scans using an intraoral reference tool. **Methods:** In this study, a custom stainless steel transfer aid carrying four 5 mm steel spheres in a trapezoidal configuration, provided with known reference distances, was used. Ten mandibular Frasaco models (in vitro) and ten healthy young adults (18–30 yrs) with intact lower arches (in vivo) received the bonded spheres. Six inter-sphere distances were defined: intermolar (BL-BR), interpremolar (FL-FR), diagonals (BL-FR, FL-BR), and lateral spans (BL-FL, BR-FR). Each arch underwent a digital scan (Medit i700) and a conventional monophase PVS impression, which was poured in Type IV stone and digitized (GOM Scan 1). The inter-sphere linear distances were measured in GOM Inspect, and trueness (deviation from reference) and precision (SD) were calculated. Data normality and homogeneity were verified; parametric *t*-tests and one-sample tests (α = 0.05) assessed differences between workflows and against reference values. **Results:** In vitro, analog impressions closely matched reference distances, with only the long-span BL-BR showing minor deviation (0.053 mm, *p* < 0.001). Digital scans showed significantly greater deviations across all spans (max 0.117 mm), particularly over long distances. In vivo, both workflows demonstrated comparable accuracy: only BL-BR (analog) and BR-FR (digital) differed significantly from the reference, and all AMEs remained within clinical thresholds (≤0.10 mm), except for BL-BR and BL-FL spans. ICC values ranged from moderate to high. Direct paired comparisons revealed statistically equivalent performance across most spans. **Conclusions:** Analog impressions outperformed digital scans in vitro, particularly across longer spans, confirming their superior dimensional fidelity under controlled conditions. However, in vivo, both workflows delivered statistically comparable and clinically acceptable accuracy. These findings suggest that while analog impressions remain the gold standard for precision-demanding contexts, modern intraoral scanners—when used correctly—can offer reliable full-arch mandibular impressions. The four-sphere reference system proved valuable for objective, anatomy-independent measurement.

## 1. Introduction

Dental impressions accurately reproduce prosthodontic field details, enabling the elaboration of various appliances. Dental impressions are a preliminary step for diagnostic and therapeutic purposes such as prosthodontics and orthodontics. The accuracy of each conventional or digital impression depends on the materials, devices, and techniques used to achieve it [1,2,3]. Thus, the impression, whether it is an intraoral optical impression or a conventional one, is a critical factor in achieving highly accurate dental restoration. Dentistry is constantly evolving, but many conventional therapeutic options remain optimal for achieving the desired aesthetic and functional results. Thus, the selection of conventional or digital methods must be adapted appropriately to each clinical situation [4].

Currently, irreversible hydrocolloids, polyethers, and polyvinylsiloxanes (PVSs) are the preferred impression materials for fixed and removable prosthodontics [5]. The dimensional stability of these materials depends on their composition, which lacks volatile secondary elements [6,7]. During clinical use, polyvinylsiloxanes have shown high accuracy in capturing intraoral anatomical details and excellent dimensional stability after setting.

Innovations in digital dental technology have revolutionized clinical and laboratory treatment workflows, making them more accessible to clinicians and significantly impacting prosthodontic treatment outcomes [8]. Digitalization aims to achieve the best results in the shortest time, at a lower cost, and with greater ease.

Intraoral scanners (IOSs) are devices with a portable digital camera (hardware), a computer, and software [9]. During digital intraoral impression, the scanner captures information about the shape and size of the dental arches, soft tissues, dental occlusion, and adjacent and antagonist teeth [10,11]. Thus, by emitting a light beam onto the tooth surface, high-resolution cameras record the distortion that the beam undergoes when it interacts with the structures of the oral cavity [12]. This distortion is processed and transformed into an accurate three-dimensional image, visible at the acquisition software interface [13]. Accuracy is one of the most essential properties of intraoral scanners. The accuracy is defined by trueness and precision [14]. Trueness refers to the accuracy with which the data generated by the scanner reflects the shape and size of the scanned anatomical elements. Good trueness means that the information obtained from the scans conforms to that from the oral cavity without errors or distortions. Precision is the ability of the scanner to reproduce the same information repeatedly during multiple scans of the same structure or region of the oral cavity [15]. Overall, the accuracy of a scan is consistent with the clinical situation and may be influenced by several factors. Accuracy depends on several factors, including user experience, scanning strategy, scanning technology, software version, moisture control (such as saliva or blood), lighting conditions, accessibility, and the dimension of the scanned area (partial/complete scan of the arch). Intraoral impressions (conventional or digital) must accurately record the details of the prosthetic field, including the exact position of the prepared teeth, the finish line, the gingival margins, and anatomical landmarks [13].

Most studies in the current literature compare digital and conventional impressions by superimposing 3D models using best-fit alignment algorithms. While informative, these methods have limitations, as the alignment process depends on the organic geometry of the dental arch and is sensitive to individual morphology and tooth positioning. As such, the introduction of systematic errors during analysis cannot be excluded, especially in the absence of a fixed physical reference.

In this context, a significant gap in the literature emerges regarding the evaluation of intraoral scanner (IOS) accuracy using objective and fixed intraoral reference objects. Unlike conventional comparison approaches, the use of physical reference systems—such as rigid bars or calibrated metal spheres—allows for precise and reproducible measurements in known spatial coordinates, independent of tooth anatomy. Recent studies by Keul et al. (2020), Güth et al. (2016), and Schlenz et al. (2022, 2023) emphasize the need for such methods in modern accuracy assessments of intraoral scanning, particularly under in vivo conditions [16,17,18,19]. However, the majority of these works remain limited to in vitro settings or focus on the maxillary arch, with minimal exploration of the mandibular arch, which presents additional challenges due to reduced anatomical stability and restricted prosthetic space.

Our study aims to evaluate the in vitro and in vivo accuracy of digital impressions compared to conventional methods in full-arch scans using an intraoral reference tool.

The null hypothesis established states that, in terms of accuracy, there are no statistically significant differences between impressions obtained by conventional and digital methods, as related to the reference values.

## 2. Materials and Methods

### 2.1. Study Design

Our study was conducted at the Faculty of Dental Medicine of George Emil Palade University of Medicine, Pharmacy, Science and Technology of Târgu Mureș to evaluate the clinical accuracy of digital impressions.

Ten patients were included in this study, taking into consideration the following inclusion and exclusion criteria:-Inclusion criteria: young patients (18–30 years), intact dental arches, healthy periodontal status, no edentulous spaces or prosthetic works, no current orthodontic treatment.-Exclusion criteria: exaggerated vomiting reflex, poor oral hygiene, malpositioned teeth, small mouth opening.

The procedures were conducted at the Department of Fixed Prosthodontics, Faculty of Dental Medicine of George Emil Palade University of Medicine, Pharmacy, Science, and Technology, of Târgu Mureș between 15 January 2024 and 15 March 2024.

The in vivo study consisted of conventional and digital impressions of the patient’s lower dental arches, captured using a transfer aid mounted on the arches.

### 2.2. Sample Size

The clinical sample size was determined based on an in vitro pilot (10 Frasaco models) modeled after Schmidt et al. (2020) and focused on the same 45 mm BL-BR intermolar span [20]. The pilot showed a mean difference of 0.07 mm between digital impressions (45.12 ± 0.12 mm) and conventional impressions (45.05 ± 0.03 mm), yielding a Cohen’s d ≈ of 0.78. This estimate aligns with the large effects reported clinically for long spans (d > 0.9) in Schmidt et al.’s study [20], providing both internal and external validation. Accordingly, a conservative d = 0.80 was entered into G*Power version 3.1.9.6 (two-tailed paired *t*-test, α = 0.05, power = 0.95), which indicated a minimum of eight paired impressions; ten patients were recruited to offset potential data loss and maintain power above 95%.

### 2.3. Transfer Aid

According to previous studies investigating the accuracy of impressions [20,21], four bearing steel spheres (1.3505, 100Cr6, DIN 5401, TIS GmbH, Gauting, Germany) with a 5 mm diameter were reversely luted on the lower arch. A transfer aid was used to position the spheres precisely.

The transfer aid was designed in collaboration with the Institute of Physics in Cluj-Napoca following a design based on the average dimensions of a dental arch. It was made of stainless steel and has four holes which, when joined together, form an isosceles trapezoid with a large base corresponding to the intermolar distance of 45 mm, a small base corresponding to the interpremolar distance of 35 mm, and sides registering a value of 24 mm (Figure 1a). Four magnetic spheres with a diameter of 5 mm were placed in these holes, representing the intraoral reference. The role of this device is to maintain the correct alignment and precise position of the spheres when they are applied to the dental arches, thereby obtaining the reference values for the distances mentioned above (Figure 1).

The transfer aid was digitized ten times using a GOM Scan 1 (ZEISS GOM GmbH, Braunschweig, Germany), a 3D scanner that utilizes fringe projection and blue light technology to capture detailed surface data of objects, thereby creating highly accurate 3D meshes. Individual scan data were aligned and merged to compute an average mesh representation of the instrument surface. Dimensional measurements were then obtained at each specified distance using GOM Inspect software (version 2018) (Figure 2).

### 2.4. In Vitro Methodology

Ten mandibular Frasaco ANA–4 typodont models (Frasaco GmbH, Tettnang, Germany) were used. The Frasaco ANA–4 typodonts used in this study include individual screw-retained teeth, which ensures high positional stability. This design minimizes any potential mobility during impression-taking or scanning procedures, eliminating a common source of bias in typodont-based in vitro studies. Thus, the risk of intra-arch distortion due to tooth mobility was considered negligible and did not require further stabilization. Four spheres were positioned on each artificial arch using a transfer aid and temporarily bonded in place with a light-cured flowable composite, allowing for reversible removal. Digital and conventional impressions of each arch were obtained. Digital scans were captured with a Medit i700 intraoral scanner, software version 2.5.2 (Medit Corp., Seoul, Republic of Korea) after applying a thin, uniform layer of scanning spray to the spheres. Conventional impressions were made using a monophase polyvinyl siloxane (PVS) impression material in standard trays, and the impressions were then poured in Type IV orthodontic dental stone to produce analog casts (Figure 3).

The analog casts were scanned with a GOM Scan 1 structured-light optical scanner (ZEISS GOM GmbH, Braunschweig, Germany). All datasets (intraoral and cast scans) were imported into GOM Inspect software for dimensional analysis. Inter-sphere distances were compared with the nominal reference values defined by the preset spacing between the magnetic spheres in the transfer aid. Meshes from the Medit i700 digital scans and the GOM Scan 1 digitization of analog casts were exported in STL format and imported into GOM Inspect software (version 2018) as mesh objects. The average reference mesh generated during device validation was imported as CAD data. On both the device CAD and each dental-arch scan mesh, geometric elements were fitted to the four spheres, and a reference line connecting the two anterior (frontal) spheres was constructed to standardize orientation (Figure 4). Alignment was carried out by best-fit registration constrained to the spherical surfaces because the Frasaco typodont teeth are not perfectly aligned and, therefore, are unsuitable as a reliable registration geometry. The spheres provide well-defined, mathematically ideal shapes that the dimensional-analysis software can consistently identify.

Spheres were labeled as back left (BL), back right (BR), front left (FL), and front right (FR). Measurements for all sides (top base, FR-FL, bottom base BR-BL, opposite sides BR-FR and BL-FL, and diagonals BR-FL and BL-FR) of the trapezoid were performed on the average reference mesh generated during device validation.

### 2.5. In Vivo Methodology

After isolating the prosthetic field with Optragate and cotton rolls, the device was tested on the lower arch. The spheres were temporarily bonded to the occlusal surfaces of the molars and premolars using light-cured flowable composite, while the transfer aid maintained consistent positioning. This approach minimized intraoral variability and ensured repeatable placement, effectively reducing alignment or measurement errors introduced by operator or anatomical inconsistencies (Figure 5).

The intraoral scanner was calibrated according to the manufacturer’s instructions to avoid deviations during scanning. Intraoral scanning was performed using the manufacturer’s recommended scanning strategy, a combined scanning strategy that involves linear movements with an occlusal starting point for posterior teeth and rotational movements for anterior teeth.

After performing the intraoral scanning, a conventional impression was made using a single-phase polyvinylsiloxane material, Zhermack Hydroise, a hydrophilic, medium-viscosity material with an automatic mixing system (Modulmix, Badia Polesine, Italy.) in a standard spoon. The intraoral setting time was 2 min and 30 s. This material has an elastic recovery of >99.5%, and the dimensional change after 24 h is <0.20%. After disinsertion, the magnetic spheres can frequently remain retained in the impression, and it is not necessary to retake the impression, as they can be easily but cautiously removed immediately after disinsertion. The cast analog models were scanned using the professional GOM SCAN 1 scanner, and the resulting scans were imported into the GOM Inspect program for measurement, employing the same technique as in the in vitro study.

The use of spheres as overlapping surfaces was employed in vivo because tooth structures are organic shapes that are difficult to assign coordinate systems to, and spheres are the only valid reference in this situation. Another reason for using this overlap is to perform measurements on different axes; this yields the most realistic and predictable results, a capability that can only be achieved through this alignment.

All intraoral scans, conventional impressions, and measurements were performed by a single experienced operator trained in both digital and conventional workflows. This approach was adopted to minimize operator-induced variability and ensure consistency throughout the study.

### 2.6. Statistical Analysis

The statistical analysis was performed using GraphPad Prism 10 for macOS version 10.2.3. Mean (M) and standard deviation (±SD) were the parameters taken into account. The statistical significance threshold was set at α = 0.05. Due to the parametric distribution of the data, confirmed by applying the Shapiro–Wilk test, and the homogeneity of variances indicated by the Levene test for statistical analysis, the following parametric tests were used for statistical analysis: paired *t*-test and one-group *t*-test.

Agreement between the two impression techniques was evaluated with unsigned accuracy metrics. For each inter-sphere span, the absolute mean error (AME), the root-mean-square error (RMSE), and the random error estimated by Dahlberg’s formula were calculated. Under previous studies, a clinically acceptable tolerance of 0.10 mm was adopted [22,23]. One-sided one-sample *t*-tests assessed whether AME stayed below this limit. Method-to-method reliability was quantified with the intraclass correlation coefficient, ICC (2,1). Bland–Altman plots were produced for visual inspection of bias and 95% limits of agreement.

## 3. Results

The results of the descriptive statistics for the in vitro measurements are shown in Table 1 and Table 2.

Table 3 shows AME ranged from 0.038 mm (BL-FL) to 0.110 mm (BL-BR). Paired *t*-tests revealed significant differences for BL-FR and BL-FL (*p* < 0.01), while other spans showed no significant deviation (*p* > 0.1). Root mean square error values followed a similar trend. Dahlberg’s formula yielded errors below 0.10 mm in all cases. Pearson correlation coefficients (ICC) ranged from −0.21 to 0.80, suggesting variable consistency depending on span length (Figure 6 and Figure 7).

*T*-tests comparing the analog model measurements with the reference values showed no significant differences across most spans, except for the BL-BR (long base) distance (discrepancy: 0.053 ± 0.033 mm). Trueness analysis indicated minimal deviations (0.001–0.053 mm), demonstrating measurements that closely match reference values and high accuracy. The highest trueness was observed for BL-BR (0.053 mm), which remains a very small value. Precision analysis shows slight standard deviations (0.033–0.093), reflecting high consistency (Table 4).

All parameters showed statistically significant differences when comparing the mean values measured on the virtual models with the reference values. Digital models exhibited greater deviations in trueness compared to analog models (0.020–0.090 mm), with the highest at BL-FL (0.090 mm). Standard deviations for precision were also slightly higher (0.038–0.181 mm), indicating increased variability (Table 5).

The results of the descriptive statistics for the in vivo measurements are shown in Table 6 and Table 7.

As shown in Table 8, absolute mean errors (AME) ranged from 0.046 mm to 0.175 mm. Except for BL-BR and BL-FL, all absolute mean errors (AMEs) were below the clinical threshold of 0.10 mm. Paired *t*-tests showed significant differences only for BL-FR and FL-FR, indicating overall comparable accuracy between the two workflows. Root-mean-square errors (RMSE) followed similar patterns. Dahlberg’s error did not exceed 0.21 mm. Pearson-based intraclass correlation coefficients (ICC) ranged from 0.26 to 0.84, reflecting moderate to high agreement (Figure 8 and Figure 9).

By applying the *t*-test to compare the means of the measurements obtained from the analog and virtual models within the patient group against reference values, it was observed that most distances showed no statistically significant differences. The exceptions were the distance at the level of the large base BL-BR of the trapezius in the analog model and the right-side BR-FR in the virtual model. Discrepancies in trueness were similar to those found in in vitro digital models (0.020–0.090 mm), with the largest at BL-FL (0.090 mm). Analyzing the standard deviation of the discrepancy (precision) revealed higher deviations than those in vitro (0.070–0.181). For in vivo analog models, slight increases in trueness deviations were noted compared to in vitro analog models (0.009–0.090), though they remained within acceptable limits, with the greatest discrepancy at the large base of the trapezium. Regarding precision, standard deviations ranged from 0.055 to 0.308 mm, with greater variability at BL-FL (0.308 mm) (Table 9 and Table 10).

## 4. Discussion

In the in vitro phase, analog impressions demonstrated superior accuracy and consistency compared to digital scans. Only one inter-sphere distance (BL-BR) in the analog workflow differed significantly from the reference (mean discrepancy = 0.053 mm, *p* < 0.001). All remaining distances showed 95% confidence intervals that included theoretical values, demonstrating negligible systematic error. The mean trueness for the analog casts was 0.019 mm (SD = 0.054 mm), which is well below the 0.1 mm threshold commonly regarded as clinically acceptable. In contrast, the digital workflow (Medit i700) yielded significant differences from nominal values across all six evaluated distances (*p* < 0.05), with a mean trueness of 0.071 mm and precision ranging from 0.038 to 0.12 mm. Notably, the most pronounced errors occurring over longer spans (BL-BR, BL-FR), consistent with findings in previous studies by Güth et al. and Schmidt et al. [17,20]. These deviations likely result from accumulated stitching errors inherent to intraoral scanning technology. Notably, ICC values in vitro ranged from poor to moderate for the digital method, supporting the observed variability. These findings suggest that while current digital scanning generally delivers acceptable accuracy for shorter spans, caution may be warranted for extended arch measurements, where accumulated deviation could become clinically relevant.

When transferred from the controlled, in vitro environment to the clinical setting, the performance profile of the two workflows shifted. Analog impressions, which had shown excellent dimensional fidelity in vitro, maintained acceptable trueness in vivo (mean = 0.054 mm) but experienced a substantial loss of precision (mean SD = 0.143 mm), as evidenced by broadened 95% confidence intervals. Statistical testing revealed only one significant deviation from the theoretical reference (BL-BR: *p* = 0.0143); all other inter-sphere distances remained statistically indistinguishable from nominal values. Even so, the considerable variability on the BL-FL span (SD = 0.308 mm) flags a potential anterior handling problem—limited visibility, saliva contamination, or tray impingement could all contribute. In contrast, the digital intraoral scans demonstrated clinically significant performance improvement. Mean trueness improved to 0.039 mm (vs 0.054 mm analog), and the mean SD narrowed to 0.125 mm. Only BR-FR reached statistical significance (*p* = 0.0253); the remaining five spans failed to reject the null hypothesis, and confidence intervals were tighter overall. Importantly, systematic discrepancies were smaller than those recorded for the digital workflow under in vitro conditions, implying that the scanner’s acquisition and software compensation routines may be tuned for the geometries and textures encountered intraorally. From a clinical standpoint, these data suggest that modern digital systems may partially mitigate environmental noise (soft-tissue dynamics, localized powdering/spray, and line-of-sight constraints) that disproportionately degrade analog impressions, particularly across extended anterior–posterior spans.

By employing absolute error metrics (AME, RMSE), ICC, and Bland–Altman plots, the study avoids the masking of systematic bias that occurs when averaging signed errors. The study also incorporated Dahlberg’s formula to estimate random error and applied one-sided one-sample *t*-tests to assess whether measurement deviations exceeded clinically acceptable thresholds (set at 0.10 mm).

In laboratory conditions, analog impressions significantly outperformed digital scans across all six measured spans. Paired *t*-tests confirmed statistically higher absolute errors in digital scans (*p* < 0.01 for all distances). The greatest discrepancies occurred in the long diagonal and posterior spans (BL-BR and BL-FR), likely due to cumulative stitching errors and geometric distortions during intraoral scanning. Conversely, analog impressions demonstrated excellent repeatability and minimal error propagation, with AME consistently below 0.060 mm.

When transitioning to clinical conditions, the differences between workflows decreased significantly. Paired comparisons showed no statistically significant difference in mean errors between digital and analog methods for most spans (*p* > 0.05), except for BL-FL and FL-FR, where analog slightly outperformed digital. Both methods achieved clinically acceptable accuracy levels (<0.10 mm) in most spans.

This convergence may result from multiple factors: intraoral humidity and limited access decrease analog accuracy, while real-time software correction and patient-specific scan strategies enhance digital performance.

Previous studies have shown that the scanning strategy can significantly influence the accuracy of the scans. This study employed the same scanning strategy for both artificial and in vivo dental arches [24]. A single operator performed the clinical and laboratory scans. The basic concept for evaluating dimensional changes in full-arch scans has been widely studied in multiple in vitro scientific papers [21]. However, few studies have examined impression accuracy using an intraoral reference. Thus, a detailed comparison with the literature is challenging due to the limited number of publications employing the same working methodology. In 2015, Vogel et al. used 5 mm diameter spheres for the first time to determine scanning accuracy in vitro [25] by scanning both conventional alginate impressions and the resulting analog models. During the in vitro determinations of the present study, significant differences were observed between the minimum and maximum values recorded for most parameters of the virtual models, particularly at long distances, compared to the analog models.

Regarding the examined average values, the lowest value was recorded at the FL-FR frontal distance, while the highest value was recorded at the BL-BR lateral distance. The most significant discrepancies were recorded over long distances. Regarding the lateral distances, the discrepancies were negative, with values lower than the reference. The discrepancies between the scanner’s accuracy and the actual differences were significant. The discrepancies in value exceeded the accuracy value but fell within the accepted clinical range of 100–120 microns. Both conventional and digital impressions have high accuracy in vivo, as there are no differences between the values measured on the models obtained by these methods. Moreover, the differences between the values obtained from the models are mainly close to the reference values for each model, with a distance that shows significant differences but falls within the clinically accepted range. In the in vivo study, the minimum and maximum differences were lower than in the in vitro study. Keul et al. conducted an in vitro study on the accuracy of conventional and digital impressions using a metal bar as an intraoral reference [16]. In most studies from the literature, better scanning results are observed for shorter distances (lateral distances) than for long intermolar and interpremolar distances [16,20], as was demonstrated in our study, with significant discrepancies over long distances (BL-BR). Ender et al. demonstrated high accuracy for conventional impressions by simultaneously achieving the entire dental arch, which explains the variables obtained in our study [26,27]. Schelnz et al. [19] studied the accuracy of conventional alginate and digital impressions on dental arches with fixed orthodontic appliances. In their study, five scanners (CS3600—Rochester, NY, USA, Primescan—Charlotte, NC, USA, Trios 4—Copenhagen, Denmark, Medit i500—Seoul, Republic of Korea, and Emerald S—Helsinki, Finland) were used with the same intraoral reference. Schlenz’s [19] results disagreed with ours, demonstrating higher accuracy when using intraoral scanners. This discrepancy may be due to the use of high-precision scanners [19].

The analysis method employed in this research, which utilizes an intraoral reference, eliminates the overlapping technique of digital models commonly used in most studies. The required digital superimposition in our study is at the level of the spheres. The deviations of the dental regions between the spheres are not recorded. The measurements are more accurate. The application of this intraoral reference enables the digital and analog datasets to be scanned with a high-precision 3D scanner, the GOM SCAN 1, compared to laboratory scanners that do not offer the same performance and have been used in multiple studies [27]. According to the aforementioned aspects, Güth et al. emphasize the avoidance of “best fit” alignment, particularly for high-information scans, such as full-arch scans, due to errors in the calculation process [17]. The scanning spray on the metallic spheres provided visible marks to the scanner and for sound image recording [28]. Ender et al. also reported good results from using scanning powder, as enamel and soft tissues have different characteristics in reflecting light, which interferes with the light projection of the intraoral scanners [26]. According to our study and Giachetti et al., conventional impressions made with high-quality materials with good dimensional stability are more accurate than optical impressions [29]. Others reported no differences between digital and conventional impression techniques [30,31,32,33]. In their study, Kontis et al. [34] used a reference metal bar positioned on the occlusal surfaces of secondary molars. They observed a reduced accuracy in models with indentations and prepared teeth [34].

The observed decrease in accuracy over long distances with intraoral scanning has important clinical implications. In restorative cases involving full-arch frameworks, long-span implant bridges, or rigid splints, even small dimensional errors can cause misfits, tension in the prosthesis, or loss of passive fit. Clinicians should be cautious about relying solely on digital impressions for these indications. Segmental scanning, verification jigs, or alternative analog workflows may be better options in high-precision cases. Our findings emphasize the importance of choosing the right scanning strategy based on the clinical complexity and the length of the arch.

Limitations of the study: As a limitation of the study, we noted the relatively small sample size and the use of a single medium-rated scanner compared to the high-precision scanners available on the market and included in the literature review. We also employed a single scanning strategy, as recommended by the manufacturer. Another limitation is the study’s focus on only the lower dental arches. According to Kuhr et al., the optical impression of the upper arch can provide better results because it also enables palate scanning, allowing the scanner software to utilize this reference for better image acquisition [21]. While typodont teeth were screw-retained, ensuring stability, the transfer jig may still introduce minor alignment variability. Potential undercuts and rounding artifacts caused by the geometry of the metal spheres and the scanner’s resolution are additional error sources. These were not directly quantified but may have contributed to slight deviations. A recent study by Nguyen et al. discusses how such rounding effects in sharp geometries can bias scanning accuracy [35]. Using custom trays instead of standard ones can improve the accuracy of conventional impressions. Additionally, the study did not include intra- or inter-operator reproducibility testing. While procedural consistency was maintained by employing a single experienced operator for all impressions, future studies should incorporate multiple clinicians and repeated trials to enhance reliability and generalizability. The study should be extended to different clinical situations (partially edentulous or protruded dental arches, dental arches with fixed prosthodontic works, and implant–prosthetic situations) by using different scanning paths and more precise intraoral scanners.

## 5. Conclusions

Within the limitations of this study, analog impressions exhibited superior dimensional accuracy under in vitro conditions, particularly in long-span measurements. However, in clinical settings, both digital and conventional techniques showed statistically similar and clinically acceptable accuracy, with most deviations remaining below the 0.10 mm threshold. Notably, intraoral scans demonstrated greater consistency in vivo, likely due to software-based real-time adjustments for intraoral variables. The use of a fixed four-sphere intraoral reference provided objective, reproducible, and anatomy-independent measurements, reducing alignment bias. While analog workflows are still preferred for high-precision cases such as full-arch rehabilitations or long-span restorations, modern intraoral scanners can serve as reliable alternatives in routine clinical practice—assuming scanning protocols are properly followed. The choice of impression technique should therefore depend on clinical complexity, span length, and the required level of precision, always considering each workflow’s specific limitations and the context of use.

## Figures and Tables

**Figure 1 diagnostics-15-02077-f001:**
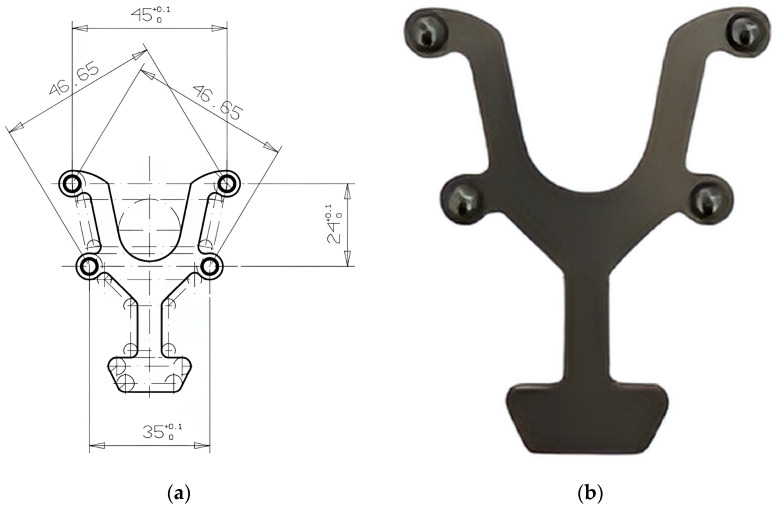
Custom transfer aid with standardized inter-sphere distances: (**a**) the trapezoid design with four magnetic spheres; (**b**) the final device with the spheres attached to establish fixed reference distances during the 3D analysis.

**Figure 2 diagnostics-15-02077-f002:**
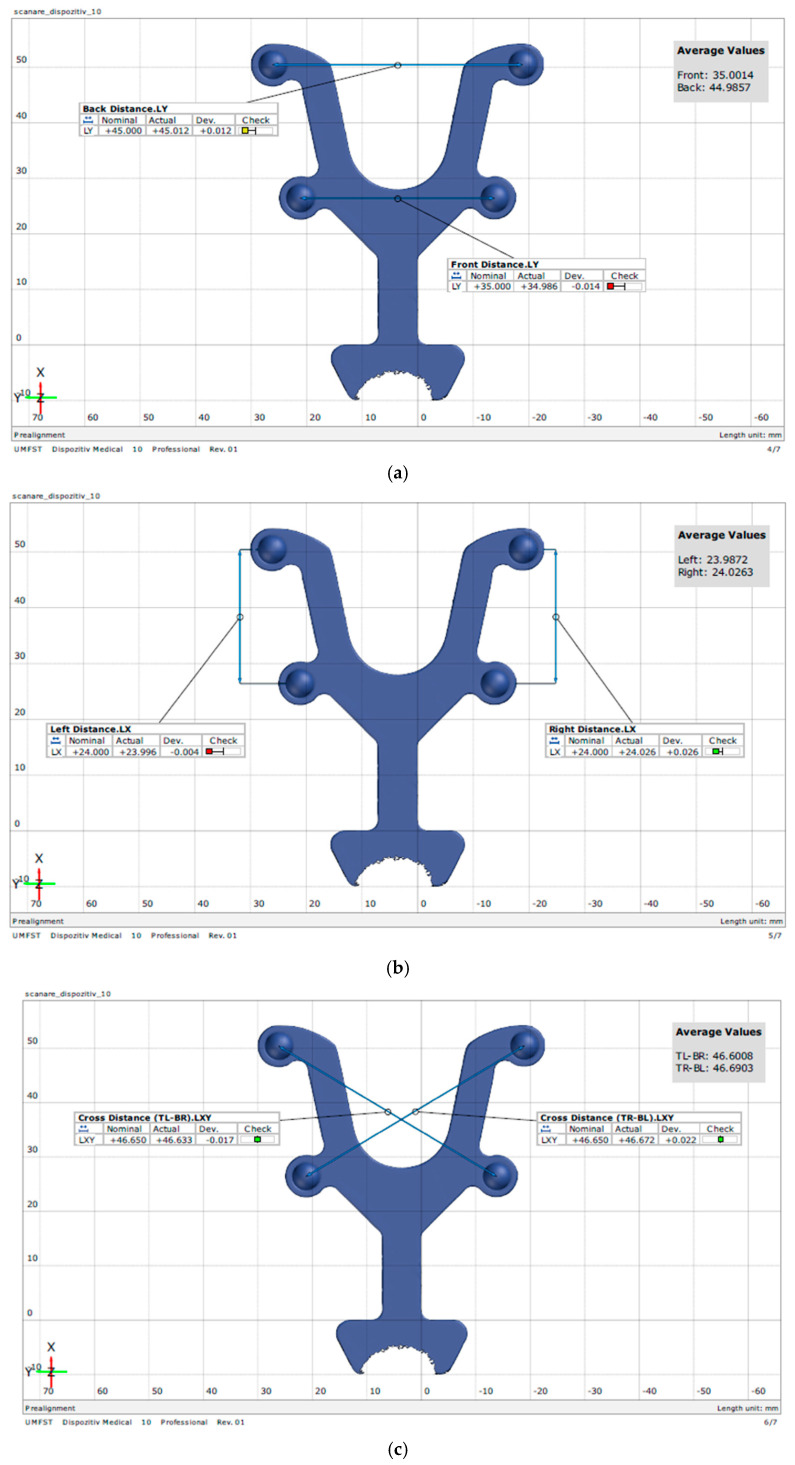
Average mesh of the transfer aid obtained in GOM Inspect software (version 2018): (**a**) distances in the *Y*–axis; (**b**) distances in the *X*–axis; (**c**) diagonal distances in the *XY*–axis.

**Figure 3 diagnostics-15-02077-f003:**
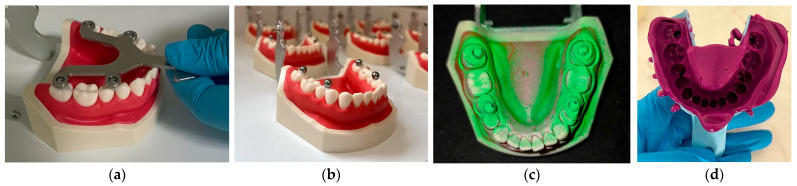
In vitro study preparation: (**a**) sphere positioning with the transfer aid; (**b**) the four spheres fixed with light-cured composite material onto the occlusal surface of the lateral teeth; (**c**) application of scanning spray before digital impression; (**d**) conventional monophase polyvinyl siloxane impression in standard tray.

**Figure 4 diagnostics-15-02077-f004:**
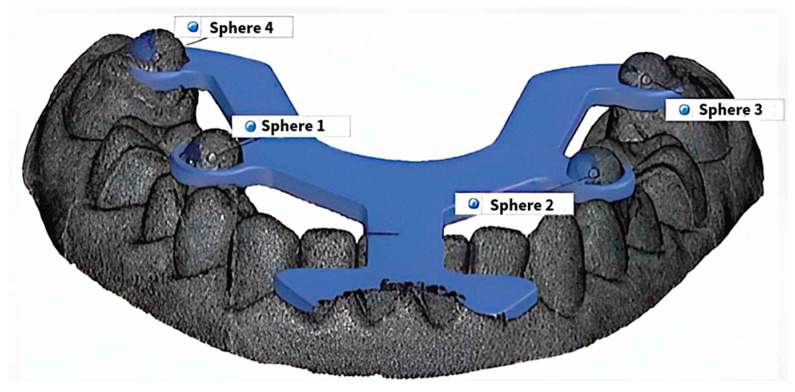
Digital alignment of the transfer aid. Overlap on the artificial arch scan mesh for standardized distances.

**Figure 5 diagnostics-15-02077-f005:**
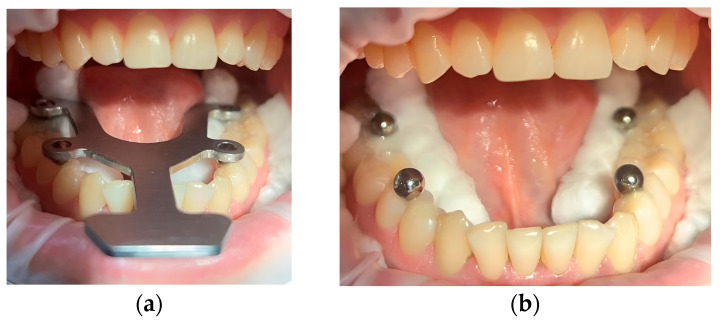
Clinical preparation for intraoral scanning procedure: (**a**) intraoral positioning of the metallic transfer aid for standardized placement of the magnetic spheres before 3D analysis; (**b**) the reference spheres temporarily luted onto the occlusal surface of the lateral teeth.

**Figure 6 diagnostics-15-02077-f006:**
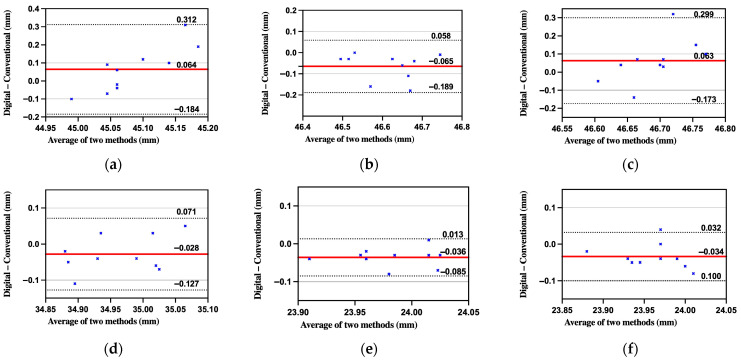
Bland–Altman agreement between in vitro digital and conventional impressions for the six inter-sphere spans: (**a**) BL-BR; (**b**) BL-FR; (**c**) FL-BR; (**d**) FL-FR; (**e**) BL-FL; (**f**) BR-FR. Solid red line = mean bias; dashed black lines = 95% limits of agreement; solid gray line = clinical threshold.

**Figure 7 diagnostics-15-02077-f007:**
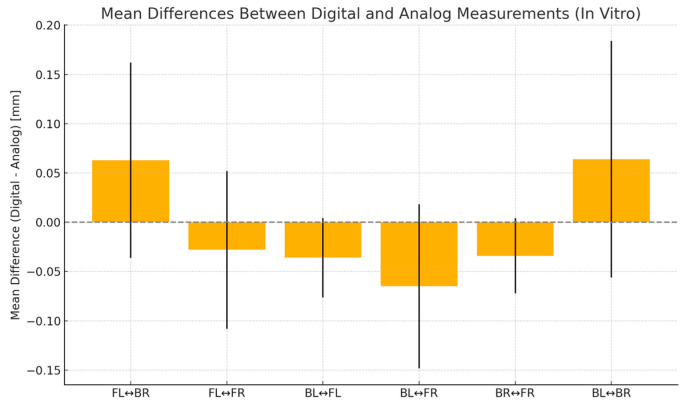
Mean differences (±SD) between in vitro digital and analog impressions for each inter-sphere span in the in vitro study. A value above the zero line indicates that digital impressions are measured longer than analog ones. The standard deviation bars indicate measurement variability.

**Figure 8 diagnostics-15-02077-f008:**
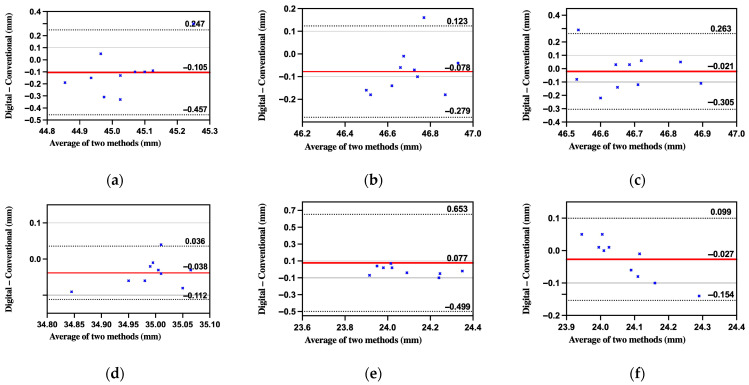
Bland-Altman agreement between in vivo digital and conventional impressions for the six inter-sphere spans: (**a**) BL-BR; (**b**) BL-FR; (**c**) FL-BR; (**d**) FL-FR; (**e**) BL-FL; (**f**) BR-FR. Solid red line = mean bias; dashed black lines = 95% limits of agreement; solid gray line = clinical threshold.

**Figure 9 diagnostics-15-02077-f009:**
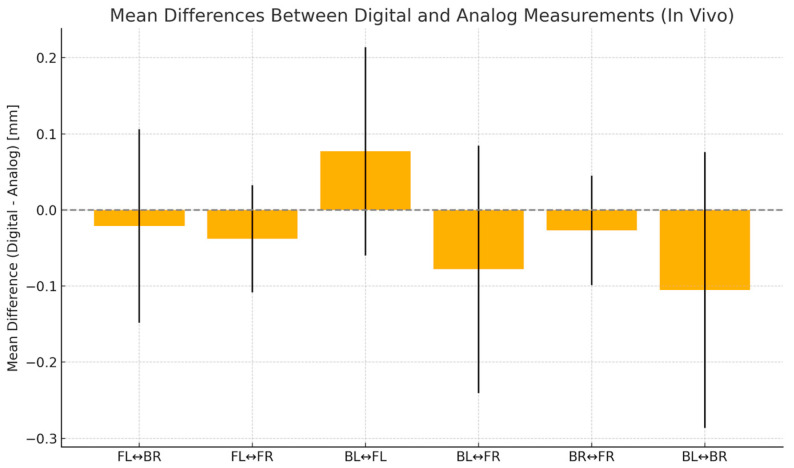
Mean differences (±SD) between in vivo digital and analog impressions for each inter-sphere span in the in vitro study. A value above the zero line indicates that digital impressions are measured longer than analog ones. The standard deviation bars indicate measurement variability.

**Table 1 diagnostics-15-02077-t001:** The descriptive statistics for in vitro analog models.

	BL-BR	BL-FR	FL-BR	FL-FR	BL-FL	BR-FR
Number of values	10	10	10	10	10	10
Minimum	45.00	46.51	46.56	34.89	23.93	23.89
Maximum	45.09	46.76	46.73	35.06	24.06	24.05
Mean	45.05	46.65	46.66	34.98	24.00	23.98
Std. Deviation	0.03335	0.09358	0.05131	0.06197	0.03900	0.04572
Lower 95% CI of mean	45.03	46.58	46.62	34.93	23.97	23.94
Upper 95% CI of mean	45.08	46.71	46.70	35.02	24.03	24.01

**Table 2 diagnostics-15-02077-t002:** The descriptive statistics for in vitro virtual models.

	BL-BR	BL-FR	FL-BR	FL-FR	BL-FL	BR-FR
Number of values	10	10	10	10	10	10
Minimum	44.94	46.48	46.58	34.84	23.89	23.87
Maximum	45.32	46.74	46.88	35.09	24.02	23.99
Mean	45.12	46.58	46.72	34.95	23.97	23.94
Std. Deviation	0.1200	0.08324	0.09913	0.08014	0.04035	0.03831
Lower 95% CI of mean	45.03	46.52	46.65	34.89	23.94	23.92
Upper 95% CI of mean	45.20	46.64	46.79	35.01	23.99	23.97

**Table 3 diagnostics-15-02077-t003:** Unsigned accuracy and random-error metrics for the six inter-sphere spans—in vitro.

Distance	Absolute Mean Error (mm)	RMSE (mm)	Dahlberg Error (mm)	Paired *t*-Test *p*	ICC
BL-BR	0.110	0.136	0.096	0.144	−0.064
BL-FR	0.065	0.088	0.062	0.010	0.751
FL-BR	0.101	0.131	0.092	0.133	−0.208
FL-FR	0.050	0.056	0.039	0.115	0.774
BL-FL	0.038	0.043	0.030	0.001	0.801
BR-FR	0.042	0.047	0.033	0.011	0.691

**Table 4 diagnostics-15-02077-t004:** Trueness and precision of the in vitro analog models.

	BL-BR	BL-FR	FL-BR	FL-FR	BL-FL	BR-FR
Theoretical mean	45.00	46.65	46.65	35.00	24.00	24.00
Actual mean	45.05	46.65	46.66	34.98	24.00	23.98
*p* value (two tailed)	0.0007	0.9215	0.5148	0.2906	0.9372	0.1461
*p* value summary	***	ns	ns	ns	ns	ns
Significant (alpha = 0.05)?	Yes	No	No	No	No	No
Discrepancy (trueness)	0.0530	−0.0030	0.01100	−0.02200	0.001000	−0.02300
SD of discrepancy (precision)	0.03335	0.09358	0.05131	0.06197	0.03900	0.04572
SEM of discrepancy	0.01055	0.02959	0.01622	0.01960	0.01233	0.01446
95% confidence interval	0.02914 to 0.07686	−0.06994 to 0.06394	−0.02570 to 0.04770	−0.06633 to 0.02233	−0.02690 to 0.02890	−0.05570 to 0.009704

***: extremely significant, ns: not significant.

**Table 5 diagnostics-15-02077-t005:** Trueness and precision of the in vitro virtual models.

	BL-BR	BL-FR	FL-BR	FL-FR	BL-FL	BR-FR
Theoretical mean	45.00	46.65	46.65	46.65	24.00	24.00
Actual mean	45.12	46.58	46.72	46.72	23.97	23.94
*p* value (two tailed)	0.0131	0.0295	0.0426	0.0426	0.0227	0.0011
*p* value summary	*	*	*	*	*	**
Significant (alpha = 0.05)?	Yes	Yes	Yes	Yes	Yes	Yes
Discrepancy (trueness)	0.1170	−0.06800	0.07400	0.07400	−0.03500	−0.05700
SD of discrepancy (precision)	0.1200	0.08324	0.09913	0.09913	0.04035	0.03831
SEM of discrepancy	0.03795	0.02632	0.03135	0.03135	0.01276	0.01212
95% confidence interval	0.03115 to 0.2028	−0.1275 to −0.008454	0.003087 to 0.1449	0.003087 to 0.1449	−0.06386 to −0.006138	−0.08441 to −0.02959

*: significant, **: very significant.

**Table 6 diagnostics-15-02077-t006:** The descriptive statistics for in vivo analog models.

	BL-BR	BL-FR	FL-BR	FL-FR	BL-FL	BR-FR
Number of values	10	10	10	10	10	10
Minimum	44.94	46.58	46.39	34.89	23.26	23.92
Maximum	45.19	46.96	46.95	35.09	24.36	24.36
Mean	45.09	46.74	46.69	35.01	24.01	24.09
Std. Deviation	0.08873	0.1287	0.1484	0.05547	0.3079	0.1312
Lower 95% CI of mean	0.02806	0.04069	0.04691	0.01754	0.09738	0.04149
Upper 95% CI of mean	45.02	46.65	46.58	34.97	23.79	23.99

**Table 7 diagnostics-15-02077-t007:** The descriptive statistics for the in vivo virtual models.

	BL-BR	BL-FR	FL-BR	FL-FR	BL-FL	BR-FR
Number of values	10	10	10	10	10	10
Minimum	44.76	46.42	46.49	34.8	23.88	23.97
Maximum	45.4	46.91	46.86	35.05	24.34	24.22
Mean	44.98	46.66	46.67	34.97	24.09	24.06
Std. Deviation	0.1814	0.1627	0.1273	0.07047	0.1368	0.07203
Lower 95% CI of mean	44.85	46.55	46.58	34.92	23.99	24.01
Upper 95% CI of mean	45.11	46.78	46.76	35.02	24.19	24.11

**Table 8 diagnostics-15-02077-t008:** Unsigned accuracy and random-error metrics for the six inter-sphere spans—in vivo.

Distance	Absolute Mean Error (mm)	RMSE (mm)	Dahlberg Error (mm)	Paired *t*-Test *p*	ICC
BL-BR	0.175	0.200	0.142	0.098	0.263
BL-FR	0.110	0.125	0.088	0.040	0.776
FL-BR	0.113	0.139	0.098	0.657	0.457
FL-FR	0.046	0.052	0.037	0.011	0.847
BL-FL	0.133	0.289	0.205	0.429	0.322
BR-FR	0.051	0.067	0.047	0.219	0.964

**Table 9 diagnostics-15-02077-t009:** Trueness and precision of the in vivo analog models.

	BL-BR	BL-FR	FL-BR	FL-FR	BL-FL	BR-FR
Theoretical mean	45.00	46.65	46.65	35	24	24
Actual mean	45.09	46.74	46.69	35.01	24.01	24.09
*p* value (two tailed)	0.0143	0.0543	0.4049	0.6202	0.8967	0.063
*p* value summary	*	ns	ns	ns	ns	ns
Significant (alpha = 0.05)?	Yes	No	No	No	No	No
Discrepancy (trueness)	0.085	0.09	0.041	0.009	0.013	0.088
SD of discrepancy (precision)	0.08873	0.1287	0.1484	0.05547	0.3079	0.1312
SEM of discrepancy	0.02806	0.04069	0.04691	0.01754	0.09738	0.04149
95% confidence interval	0.02153 to 0.1485	−0.002044 to 0.1820	−0.06513 to 0.1471	−0.03068 to 0.04868	−0.2073 to 0.2333	−0.005867 to 0.1819

*: significant, ns: not significant.

**Table 10 diagnostics-15-02077-t010:** Trueness and precision of the in vivo virtual models.

	BL-BR	BL-FR	FL-BR	FL-FR	BL-FL	BR-FR
Theoretical mean	45	46.65	46.65	35	24	24
Actual mean	44.98	46.66	46.67	34.97	24.09	24.06
*p* value (two tailed)	0.7354	0.8208	0.6312	0.2254	0.0672	0.0253
*p* value summary	ns	ns	ns	ns	ns	*
Significant (alpha = 0.05)?	No	No	No	No	No	Yes
Discrepancy (trueness)	−0.02	0.012	0.02	−0.029	0.09	0.061
SD of discrepancy (precision)	0.1814	0.1627	0.1273	0.07047	0.1368	0.07203
SEM of discrepancy	0.05737	0.05146	0.04025	0.02228	0.04326	0.02278
95% confidence interval	−0.1498 to 0.1098	−0.1044 to 0.1284	−0.07105 to 0.1111	−0.07941 to 0.02141	−0.007853 to 0.1879	0.009476 to 0.1125

*: significant, ns: not significant.

## Data Availability

The dataset analyzed during this study are available from the first author on request.

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
