# Peer review of "Accuracy of Analog and Digital Full-Arch Mandibular Impressions: In Vitro and In Vivo Evaluation"

_diagnostics, 2025, doi:10.3390/diagnostics15162077_

Round 1
Reviewer 1 Report
Comments and Suggestions for Authors
-
Introduction: The background lacks depth and fails to identify the gap in the literature clearly. Please provide a more structured justification for focusing on full-arch mandibular impressions and the use of intraoral reference objects. Many cited studies are outdated or lack critical analysis.
-
Study design: Although the dual-phase (in vitro and in vivo) design is commendable, the sample size is small (n = 10), and no assessment of intra- or inter-operator reliability was conducted. This significantly limits the generalizability and statistical strength of the findings.
-
Methods: The methodology is described in good detail. However, the authors should explain how the bonding of the spheres (in vivo) was standardized and whether the intraoral positioning variability could affect the measurement.
-
Results: The results section is too verbose, and the figures are not well formatted. Several graphs lack axis labels and clarity. Consider summarizing key statistics and emphasizing only clinically significant findings.
-
Discussion: The discussion is overly general and not sufficiently critical. The authors should better contrast their findings with recent high-quality studies using intraoral reference bars or coordinate-based systems, such as those by Keul, Schlenz, or Guth.
-
Figures and Tables: Many figures lack proper legends or have inconsistent formatting. Figures 5–8 are difficult to interpret without clarification. Ensure that all visuals meet the standards of scientific presentation.
-
Conclusions: The conclusions seem overstated given the limitations. The authors should include a clearer discussion of the clinical implications, especially regarding long-span scanning errors.
The manuscript requires substantial language editing. Many sentences are long, imprecise, or grammatically incorrect. There is excessive use of vague or subjective phrases (e.g., “instilling confidence”), which detracts from scientific clarity. Please consider having the manuscript professionally edited by a native English speaker with experience in academic or biomedical writing. Rewriting for conciseness, clarity, and grammatical accuracy is strongly recommended.
Reviewer 2 Report
Comments and Suggestions for Authors
The manuscript tackles an important clinical question with an innovative four‑sphere reference device, but its statistical approach and reporting require revision.
Undefined abbreviations. The first occurrence of BL‑BR (and the other sphere labels) appears in the Abstract without explanation, which will confuse readers unfamiliar with the jig.
Sample‑size justification. An a‑priori effect size of 0.80 is very large. Unless it was derived from pilot data or a published study, the manuscript should justify this value explicitly or adopt a more conservative “medium” default.
Potential mobility of typodont teeth. Even slight play in the typodont can bias both impressions and scans. The manuscript should describe how the teeth were rigidly stabilised or acknowledge this limitation and discuss its impact on accuracy.
Choice of accuracy metric and statistics. Averaging signed errors allows positive and negative deviations to cancel out, hiding systematic bias. Re‑analyse the data using absolute mean error or root‑mean‑square error, estimate random error with Dahlberg’s formula, and illustrate agreement with Bland‑Altman plots. A one‑sided one‑sample t‑test against a clinically acceptable threshold will then show whether the method is accurate enough. Also, Intraclass Correlation coeffecient should be reported for reliability.
No direct comparison between workflows. Judging each workflow against the reference does not reveal which one is more accurate. Once absolute errors are calculated, compare digital and PVS impressions directly with paired or matched tests to fulfil the study’s stated aim.
Scanner model. The Medit i700 is no longer the newest hardware; the i900 is now current. Please state the scanner’s firmware/software version, explain why the i700 was chosen, and acknowledge that newer hardware could produce different results.
Undercut and rounding effects. Accuracy losses caused by under‑cuts beneath the spheres and by rounding of sharp edges on the jig are not discussed. Including these sources of error and citing recent work on similar phenomena would strengthen the Discussion.
Cite related articles doi:10.1177/17436753241277332 on rounding artefacts in intra‑oral scanning and provide directly relevant context.
Reviewer 3 Report
Comments and Suggestions for Authors
This study aimed to evaluate the in vitro and in vivo accuracy of digital impressions compared to conventional methods in full-arch scans using an intraoral reference tool. Ten mandibular typodonts (in vitro) and ten healthy young adults were included. Despite some minor discrepancy, both methods were comparable. This is a solid study and deserves publication. However some comments as follows:
1. The entire article may still needs scientific editing.
Eg. “dimensionally faithful" is not a scientific writing.
2. Please discuss the factors that might impact the accuracy in-vivo and in-vitro. Ideally with comparable referenced studies.
3. Fig 7. Please highlight some major difference in text and potentially summarize in the abstract.
4. "BL-BR" how can the author translate its meaning to general pracicianers.
5. Please discuss the potential error from the measuring device
Round 2
Reviewer 2 Report
Comments and Suggestions for Authors
The manuscript has been thoroughly revised. I suggest acceptance in the present form.